# Knee Osteoarthritis Progression Is Delayed in Silent Information Regulator 2 Ortholog 1 Knock-in Mice

**DOI:** 10.3390/ijms221910685

**Published:** 2021-10-01

**Authors:** Tetsuya Yamamoto, Nobuaki Miyaji, Kiminari Kataoka, Kyohei Nishida, Kanto Nagai, Noriyuki Kanzaki, Yuichi Hoshino, Ryosuke Kuroda, Takehiko Matsushita

**Affiliations:** Department of Orthopaedic Surgery, Kobe University Graduate School of Medicine, Kobe 650-0017, Japan; ytetsu36@gmail.com (T.Y.); nobuakimiyaji@gmail.com (N.M.); k.s.kimi.h.m.m.cp@gmail.com (K.K.); kyohei_nishida0610@yahoo.co.jp (K.N.); kantona9@gmail.com (K.N.); kanzaki@med.kobe-u.ac.jp (N.K.); you.1.hoshino@gmail.com (Y.H.); kurodar@med.kobe-u.ac.jp (R.K.)

**Keywords:** silent information regulator 2 ortholog 1 (SIRT1), osteoarthritis, knock-in mice, knee

## Abstract

Overexpression of silent information regulator 2 ortholog 1 (SIRT1) is associated with beneficial roles in aging-related diseases; however, the effects of SIRT1 overexpression on osteoarthritis (OA) progression have not yet been studied. The aim of this study was to investigate OA progression in SIRT1-KI mice using a mouse OA model. OA was induced via destabilization of the medial meniscus using 12-week-old SIRT1-KI and wild type (control) mice. OA progression was evaluated histologically based on the Osteoarthritis Research Society International (OARSI) score at 4, 8, 12, and 16 weeks after surgery. The production of SIRT1, type II collagen, MMP-13, ADAMTS-5, cleaved caspase 3, Poly (ADP-ribose) polymerase (PARP) p85, acetylated NF-κB p65, interleukin 1 beta (IL-1β), and IL-6 was examined via immunostaining. The OARSI scores were significantly lower in SIRT1-KI mice than those in control mice at 8, 12, and 16 weeks after surgery. The proportion of SIRT1 and type II collagen-positive-chondrocytes was significantly higher in SIRT1-KI mice than that in control mice. Moreover, the proportion of MMP-13-, ADAMTS-5-, cleaved caspase 3-, PARP p85-, acetylated NF-κB p65-, IL-1β-, and IL-6-positive chondrocytes was significantly lower in SIRT1-KI mice than that in control mice. The mechanically induced OA progression was delayed in SIRT1-KI mice compared to that in control mice. Therefore, overexpression of SIRT1 may represent a mechanism for delaying OA progression.

## 1. Introduction

Osteoarthritis (OA) affecting the knee joint is one of the most common joint diseases and causes joint pain and disability. Although OA has a multifactorial etiology and eventually affects the entire joint, its central pathological feature involves the progressive loss of articular cartilage [1,2]. Treatment using disease-modifying OA drugs (DMOADs) improves the underlying OA pathophysiology, thereby inhibiting structural damage to prevent or reduce long-term disability and offer potential symptomatic relief [3]. Currently, there are no effective DMOADs available that are suitable for treatment when the cartilage has been lost [2].

Sirtuins are members of the class III histone deacetylase family and regulate diverse cellular activities in aging [4]. Silent information regulator 2 type 1 (SIRT1) is sirtuin homolog and regulates various vital signaling pathways such as DNA repair and apoptosis, myogenic and adipogenic differentiation, mitochondrial biogenesis, and glucose and insulin homeostasis [5]. Studies have shown that regulation of SIRT1 may affect OA progression. In human chondrocytes, SIRT1 inhibits apoptosis and promotes cartilage-specific gene expression [6,7], and SIRT1 inhibition regulates the expression of genes related to OA [8,9]. Additionally, several genetic Sirt1-deficient mouse models exhibit accelerated OA progression [10,11,12]. Therefore, SIRT1 may play a role in protecting chondrocytes and preventing OA development.

Resveratrol (3,4,5-trihydroxystlben; RSV) is a natural product isolated from most grape cultivars; it can activate SIRT1 [4]. New molecules were identified that can stimulate sirtuin activities to a greater extent than that obtained using RSV, such as SRT1720, SRT2104, SRT2379, and other molecules [13]. We previously found that the intraperitoneal (i.p.) injection of SRT1720 reduces OA progression in a mouse model [14]. Additionally, both i.p. and intra-articular injections of SRT2104 reduce OA progression in an OA mouse model [15]. The findings of these studies using natural compounds and chemical activators strongly suggest that activation of SIRT1 can inhibit OA progression. However, the direct effect of overexpression of SIRT1 on OA suppression has not yet been investigated. 

SIRT1 transgenic mice have been generated in which the SIRT1 cDNA has been knocked into the β-actin locus (SIRT1-KI) [16]. SIRT1-KI mice are characterized as lean mice showing low levels of blood cholesterol, high glucose tolerance, and active metabolism [16]. In previous studies on SIRT1-KI mice, the stiffness of the aorta has been found to be suppressed [17] and the reproductive capacity is prolonged [18]. These studies suggest that SIRT1 overexpression plays beneficial roles in aging-related diseases and SIRT1-KI mice represent useful models to examine the effects of SIRT1 overexpression on aging-related diseases. Although previous studies using chemical SIRT1 activators and natural compounds of SIRT1 activators have suggested beneficial roles of SIRT1 on OA progression, non-direct effects of those activators have been also reported [19]. In addition, effects of constitutive overexpression of SIRT1, which could have potential adverse effects of OA progression, have not yet been examined. Therefore, the aim of this study was to investigate whether OA progression is suppressed in SIRT1-KI mice with OA to examine the effects of SIRT1 overexpression in vivo and to evaluate the possible mechanisms associated with delayed progression.

## 2. Results

### 2.1. Features of SIRT1-KI Mice

SIRT1-KI mice were maintained by mating heterozygous SIRT1-KI mice with wild type littermates. SIRT1-KI mice were identified via genotyping. The proportion of SIRT-KI mice was approximately 40% (one to three out of four to six newborn pups) (Figure 1A). During postnatal growth, SIRT1-KI mice showed no significant differences in skeletal and body weight compared to that of the littermate control mice (up to postnatal 12 months), although there was a tendency that the development of ossification was slightly delayed in SRIT1-KI mice compared with wild type mice (Figure 1B–D). Sirt1 mRNA expression was significantly higher in SIRT1-KI mice than in control mice. In articular cartilage samples, Sirt1 was significantly upregulated compared to that in muscle tissues, and there was no significant decrease in SIRT1 expression with growth (Figure 1E). Safranin-O fast staining and hematoxylin-eosin staining of control and SIRT1-KI mouse tissues at 3 weeks of age are shown in Figure 1F.

### 2.2. Reduced Severity of Cartilage Loss in SIRT-KI Mice after Destabilization via Medial Meniscus Surgery

Destabilization of the medial meniscus surgery was performed in the knee joint of 12-week-old mice. The knees of control mice and SIRT1-KI mice were collected at 4, 8, 12, and 16 weeks postoperatively. Histological analysis showed that OA developed gradually in all groups. The OARSI scores of the medial femoral condyle and tibial condyle in the control group were significantly higher than those in the SIRT-KI group at 8, 12, and 16 weeks, but not at 4 weeks (Figure 2).

### 2.3. SIRT-KI Mice Showed Lower Levels of Cartilage Matrix-Degrading Enzymes, Apoptosis, and Inflammatory Cytokines than Those in Control Mice after OA Induction via Destabilization of Medial Meniscus

Immunohistochemical analyses revealed that a considerably higher number of SIRT1-positive chondrocytes were detected in the SIRT1-KI group than that in the control group at 8, 12, and 16 weeks after surgery (Figure 3). The proportion of type-II collagen-positive chondrocytes significantly increased in the SIRT-KI group compared to that in the control group at 8 weeks after surgery. The proportions of MMP-13-, ADAMTS-5-, cleaved caspase 3 (apoptotic marker)-, PARP p85 fragment (apoptotic marker)-, NF-κB P65 (inflammatory regulator)-, IL-1β-, and IL-6-positive chondrocytes significantly decreased in the SIRT1-KI group compared to that in the control group at 8 weeks after surgery (Figure 4).

### 2.4. SIRT1-KI Chondrocytes Showed Higher Expression of Extracellular Matrix Genes than That in the Control Group, whereas Cartilage Degrading Enzyme Genes Were Downregulated

Real-time PCR analysis showed that the mRNA expression of Sirt1, Col2a1, and Acan was significantly higher and Col10a1 expression was lower in chondrocytes of SIRT1-KI mice compared to those in control mouse chondrocytes. Stimulation with IL-1β significantly reduced Sirt1, Col2a1, and Acan mRNA expression in chondrocytes of both SIT1-KI and control mice; however, expression of those genes was significantly higher in chondrocytes of SIT1-KI mice compared to that in control mice.

The stimulation with IL-1β significantly increased Mmp-3, Mmp-13, and Adamts-5 mRNA expression in both chondrocytes of both mouse groups; however, the upregulation was attenuated in chondrocytes of SIRT1-KI mice compared to that in control mice (Figure 5).

### 2.5. Microarray Analysis Identified 21 Genes Associated with SIRT1 Overexpression

Microarray analysis was performed to identify the genes associated with and regulated by SIRT1 overexpression. In total, 75 genes were found to be upregulated by more than two-fold whereas 45 genes were downregulated to <50% of their original expression level, without IL-1β stimulation in SIRT1-KI chondrocytes compared to that in control mouse chondrocytes (Figure 6A). Moreover, under stimulation with IL-1β, 119 genes were found to be upregulated by more than two-fold and 104 genes were downregulated to <50% in SIRT1-KI chondrocytes compared to that in control chondrocytes (Figure 6B). The expression of 11 genes was consistently two-fold higher and that of 10 genes was lower in SIRT-KI chondrocytes compared to that in control chondrocytes (Figure 6C) regardless of IL-1β stimulation (Table 1). Among those genes, XIST was found to be consistently downregulated in SIRT1-KI chondrocytes compared with wild type mice (Figure 7).

## 3. Discussion

The effects of SIRT1 overexpression on OA progression have not yet been investigated in SIRT-KI mice. The main finding of this study was that OA progression was delayed in SIRT1-KI mice compared to that in control mice in an experimental OA model. The delayed OA progression in SIRT1-KI mice was associated with an increased number of type II collagen and SIRT1-positive chondrocytes and decreased number of MMP-13-, ADAMTS-5-, cleaved caspase 3-, PARP p85 fragment-, acetylated NF-κB P65-, IL-1β-, and IL-6-positive chondrocytes compared to that in control mice. Additionally, the expression of extracellular matrix genes was higher in SIRT1-KI chondrocytes than that in control mice regardless of stimulation with IL-1β. Further, the upregulation of MMP-3, MMP-13, and ADMTS5 via stimulation with IL-1β was reduced in SIRT1-KI chondrocytes compared to that in control chondrocytes.

In early studies, the role of SIRT1 in the regulation of apoptosis via deacetylation of P53 has been highlighted as a main functional role of SIRT1 [20]. Thereafter, protective roles of SIRT1 against apoptosis in chondrocytes via various pathways were also demonstrated in previous studies [6,7,21]. Therefore, the reduced number of apoptotic chondrocytes in SIRT1-KI mice during OA development might have partially contributed to the delayed OA progression in SIRT-KI mice.

Regarding regulation of cartilage extracellular matrix gene expression changes by SIRT1, it has been reported that SIRT1 induces Col2a1 and Acan expression via an interaction with SOX9 in human chondrocytes [8,22,23]. Similarly, our previous study showed that the SIRT1 activator, SRT2104, stimulates Col2a1 expression in mouse chondrocytes [15]. The findings of these studies strongly suggest that SIRT1 functions as a positive regulator for extracellular matrix genes. In this study, Col2a1 and Acan expression was increased in SIRT1-KI chondrocytes regardless of IL-1β stimulation. The results of the present study agree with those of previous studies and suggest that the upregulation of extracellular matrix genes via overexpression of SIRT1 may represent a mechanism of delayed OA progression in SIRT1-KI mice. Regarding SIRT1 expression, it was reduced by the treatment with IL-β in SIRT-KI chondrocytes and SIRT1-KI mice during development of OA, although the expression level was still higher than in control chondrocytes and mice. Since SIRT1 was overexpressed under the β-actin promoter, it is possible that IL-1β reduced SIRT1 expression by decreasing promoter activity. Of interest, Yurube et al. reported that β-actin expression was reduced during the development of disc degeneration [24] Therefore, β-actin expression may be susceptible to cellular stresses and the SIRT1 expression regulated by the β-actin promoter reduced even in SIRT-KI chondrocytes and mice, although detailed mechanism needs to be examined.

The expression of acetylated NF-κB p65 subunit and cartilage-degrading enzymes, including MMP-3, MMP-13, and ADAMTS-5, was decreased in the SIRT1-KI group compared to that in the control group. Additionally, the expression of IL-1β and IL-6 was decreased in the SIRT1-KI group compared to that in the control group. These observations are consistent with our previous findings: the proportion of MMP-13- and acetylated NF-κB p65-positive chondrocytes is decreased in mice treated with SIRT1 activators [14,15], and increased in cartilage-specific Sirt1-knockout mice12 compared to that in control mice. Previous studies have shown that the NF-κB pathway mediates the inflammatory responses of chondrocytes [25,26,27] and expression of inflammation-related genes, including MMP-3 and MPP-13, IL-6, IL-1, tumor necrosis factor alpha [28]. Yeung et al. have demonstrated that SIRT1 suppresses NF-κB signaling via deacetylation of the NF-κB p65 subunit [29]. Moreover, we found that the overexpression of SIRT1 in chondrocytes decreases the expression of acetylated NF-κB p65 and MMPs induced via IL-1β stimulation in vitro [30]. Therefore, OA development in the SIRT-KI group might have been attenuated via downregulation of cartilage-degrading enzymes in chondrocytes via modulation of the NF-κB pathway. Moreover, the involvement of additional pathways in the regulation of MMP expression by SIRT1 has been suggested, such as LEF1-dependent regulation [31] and p38, JNK, and ERK phosphorylation in chondrocytes [32]. Therefore, it is also possible that the reduction in levels of cartilage-degrading enzymes in SIRT-KI mice is associated with other mechanisms and pathways.

SIRT1 plays protective roles in chondrocytes; however, SIRT1 also has been reported to play important roles in the regulation of inflammatory responses in macrophages. Park et al. reported that SIRT1 activation by resveratrol suppresses the lipopolysaccharide/interferon γ-induced NF-κB activity in macrophages in rheumatoid arthritis, and inflammatory M1 polarization is reduced in SIRT1 transgenic mice. Further, collagen-induced arthritis is attenuated in SIRT1-transgenic mice associated with an increase level of M2 macrophage markers [33]. We have also found that delayed OA progression in mice treated with SRT2014 is associated with reduced M1 and increased M2 macrophage marker levels in the synovium [15]. Although the effects of SIRT1 overexpression were not examined in this study, modulation of macrophage polarization in the synovium may have also contributed to the delayed OA progression in SIRT1-KI mice.

Microarray analysis showed that 11 genes were consistently upregulated by more than two-fold while 10 genes were downregulated to less than 50% of their original expression level in SIRT-KI chondrocytes compared to those in control chondrocytes regardless of IL-1β stimulation. Among those genes, XIST was found to be consistently downregulated in SIRT1-KI chondrocytes. Recently, many studies have demonstrated important roles of lncRNA and miRNAs in the process of OA [34,35]. XIST, a lncRNA regulating X-linked chromosomal inactivation, has been recently suggested to be involved in pathogenesis of OA. The expression of XIST was increased in OA cartilage and knockdown of XIST improved the inflammatory microenvironment in OA via acting on M1 macrophages [36]. Further, XIST knockdown ameliorated IL-1β-induced decreased cell viability and downregulation of COL2A1 and aggrecan [37]. Therefore, XIST might play a role in the delayed OA progression in SIRT1-KI mice. However, further studies are required to elucidate the role of XIST in the SIRT-KI mice.

This study has a few limitations. First, the mean body weight of mice in the control group tended to be higher than that in the SIRT1-KI group, and the difference in body weight might have affected OA progression. However, these differences were not significant; therefore, the effect might have been negligible. Second, only female mice were used in this study and different results may be obtained using male mice. Third, OA progression was examined in a posttraumatic OA model, and the features of OA progression during aging were not studied. Finally, since sham control was not included, we cannot rule out the possibility that the upregulation of cartilage anabolism and downregulation of catabolism were stimulated by the OA induction although those responses were generally opposite to those during OA progression.

## 4. Materials and Methods

### 4.1. Mice

Sirt1-overexpressing (Sirt1; C57BL/6-Actbtm3.1 (Sirt1) Npa/J) mice (The Jackson Laboratory, Bar Harbor, ME, USA) were used in this study. Mice were maintained under pathogen-free conditions and were allowed free access to food, water, and activity. All animal experiments were performed according to protocols approved by the Institutional Animal Care and Use Committee at Kobe University Graduate School of Medicine (approval number P150604), Kobe, City.

### 4.2. Genotyping

SIRT1-KI mice were maintained by mating heterozygous SIRT1-KI mice with wild type littermates. DNA was extracted from mouse tails using DNeasy Blood & Tissue Kit (Qiagen, Valencia, CA, USA) according to the manufacturer’s instructions. Polymerase chain reaction (PCR) was performed using three primers: 10,967 primer, 5′-TATGGAATCCTGTGGCATCCATGA-3′; 10,968 primer, 5′-CAAAGCCATGCCAATGTTGTCTCT-3′; 10,969 primer, 5′-GGCACATGCCAGAGTCCAAGTTTA-3′. PCR products were separated on a 2% agarose gel and visualized via ethidium bromide staining. Genotyping was performed as follows: 257 bp for SIRT1-KI type; and 547 bp for wild type.

### 4.3. Skeletal Preparation and Body Weight Analysis

Postnatal day 2 SIRT1-KI mice and control littermates were skinned, eviscerated, and fixed in 95% ethanol. The tissues were placed in a mixture of acetic acid: 95% ethanol (1:4) for 3 h to lower the pH. Alcian Blue staining was performed and the tissues were placed in potassium hydroxide to remove the soft tissue. After visualizing the cartilage via staining, Alizarin Red staining was performed. The length of long bones and vertebra (first to fifth lumbar spines) of control and SIRT1-KI littermate embryos at postnatal day 2 was measured using a divider. The weight of the mice was measured every month until one year of age.

### 4.4. Sirt1 Expression in Control and SIRT1-KI Mice

The expression of Sirt1 in muscle (tibialis anterior and quadriceps muscle) and cartilage was examined in 7-day-old and 3-month-old control and SIRT1-KI mice respectively. RNA was isolated using RNeasy Kit (Qiagen), and cDNA was synthesized using High-Capacity cDNA Reverse Transcription Kits (Applied Biosystems, Waltham, MA, USA). Real-time PCR analysis was performed using TaqMan assays (Applied Biosystems) to determine the expression of Sirt1 in duplicate for each sample, and the relative gene-expression levels were normalized to glyceraldehyde 3-phosphate dehydrogenase (Gapdh) expression levels as a reference control, based on the comparative cycle-threshold method.

### 4.5. OA Model

Based on our previous study [15], female mice were anaesthetized via i.p. injection containing a combination anesthetic (0.3 mg/kg of medetomidine, 4.0 mg/kg of midazolam, and 5.0 mg/kg of butorphanol), and the knee joint was exposed via the medial parapatellar approach. Experimental OA was induced in the knee joint of 12-week-old mice by resecting the medial meniscotibial ligament under a microscope to destabilize the medial meniscus [38]. The joint capsule and skin were closed using 3-0 nylon sutures.

### 4.6. Cell Culture and Real-Time PCR Analysis

Articular cartilage samples were collected from each knee joint of postnatal day 7 control and SIRT1-KI mice. The epiphyseal chondrocytes were cultured in monolayers in 6-well plates (2 × 105 cells/well) for 48 h. After reaching 80% confluency, the chondrocytes were incubated for 24 h with or without 10 ng/mL interleukin 1 beta (IL-1β) (R&D Systems, Minneapolis, MN, USA). The concentration of IL-1β was chosen based on the previous study [15]. RNA was isolated using RNeasy Kit (Qiagen), and cDNA was synthesized using High-Capacity cDNA Reverse Transcription Kits (Applied Biosystems). Real-time PCR analysis was performed using TaqMan assays to analyze the expression of Sirt1, Col2a1, Col10a1, Mmp-3, Mmp-13, Acan, Adamts-5, and Xist (Applied Biosystems) in duplicate for each sample, and the relative gene-expression levels were normalized to Gapdh expression levels as a reference control, based on the comparative cycle-threshold method (Table 2).

### 4.7. Histological Analysis

The control and SIRT1-KI mice were euthanized at 3 weeks of age and at 4, 8, 12, and 16 weeks after surgery (*n* = 5 mice/group for each time point). The entire knee joints were fixed in 4% paraformaldehyde in 0.1 M phosphate-buffered saline overnight at 4 °C, decalcified for two weeks using 10% ethylenediaminetetraacetic acid, and embedded in paraffin wax. Each specimen was cut into 6 μm slices along the sagittal plane and stained with safranin O-fast green and hematoxylin-eosin. Three slices were selected from each medial femoral condyle and medial tibial plateau, and photographs were taken at a magnification of 40×. The histological OA grade for each field was evaluated using the Osteoarthritis Research Society International (OARSI) cartilage OA histopathology grading system (score 0 to 6) [39]. OA grading was assessed by a single observer (NM) who was blinded to the study groups.

### 4.8. Immunohistochemistry

Deparaffinized sections were digested using proteinase (Dako Denmark AS, Glostrup, Denmark) for 10 min and treated with 3% hydrogen peroxide (Wako Pure Chemical Industries Ltd., Osaka, Japan) to block endogenous peroxidase activity. After epitope retrieval, the sections were incubated overnight at 4 °C with primary antibodies against the following mouse proteins: SIRT1 (1:50, Millipore, Billerica, MA, USA), type-II collagen (1:100, Abcam, Cambridge, UK), MMP-13 (1:100, Abcam), ADAMTS-5 (1:100, Santa Cruz Biotechnology Inc., Santa Cruz, CA, USA), cleaved caspase-3 (1: 100, Cell Signaling Technology, Tokyo, Japan), poly(ADP-ribose) polymerase (PARP) p85 (1:100, Promega, Madison, WI, USA), acetylated NF-κB p65 (1: 100, Sigma-Aldrich, St. Louis, MO, USA), IL-1β (1:100, Abcam), and IL-6 (1:100, Abcam). The anti-SIRT1 antibody was a mouse monoclonal antibody, and the others were rabbit polyclonal antibodies. Sections were then incubated with peroxidase-labeled anti-rabbit or mouse immunoglobulin (Histofine Simple Stain MAX Po; Nichirei Bioscience, Tokyo, Japan) at room temperature (25–28 °C) for 30 min, and the signals were developed as brown reaction products using the peroxidase substrate, 3,3-diaminobenzidine, with methyl green or hematoxylin counterstaining. As negative controls, a non-immune mouse or rabbit IgG (1:50 dilution) was used instead of the primary antibodies. SIRT1 expression was evaluated at the age of 3 weeks and at 4, 8, 12, and 16 weeks after surgery, and the expression of other proteins was evaluated at 8 weeks after surgery. All images were obtained under a microscope (Biozero; Keyence Corp., Itasca, OH, USA).

### 4.9. Microarray Analysis

The articular cartilage was collected from each knee joint of postnatal day 7 control wild-type littermates and SIRT1-KI mice. The epiphyseal chondrocytes were cultured in monolayers in 6-well plates (2 × 105 cells/well) for 48 h. After reaching 80% confluency, the chondrocytes were incubated for 24 h with or without 10 ng/mL IL-1β (R&D Systems). RNA was isolated using RNeasy Kit (Qiagen). After performing total RNA extraction, the samples were submitted to Kurabo Industries Ltd. (Okayama, Japan). An Affymetrix GeneChipTM Mouse Gene 2.0 ST Array was used to compare expression data between control and SIRT1-KI mice.

### 4.10. Statistical Analysis

An unpaired two-tailed Student’s *t*-test was performed to compare differences between groups. One-way analysis of variance was performed to compare multiple groups using the Bonferroni method as a post-hoc test; *p*-values < 0.05 were considered significant. Data were analyzed using BellCurve for Excel (Social Survey Research Information Co., Ltd., Tokyo, Japan).

## 5. Conclusions

We have demonstrated that OA progression was delayed in the SIRT1-overexpressing SIRT1-KI mice in an experimental OA model. The delayed OA progression was associated with reduction in levels of cartilage-degrading enzymes, apoptotic markers, and acetylated NF-κB p65 in chondrocytes. SIRT1 overexpression may represent a promising strategy for treatment of OA.

## Figures and Tables

**Figure 1 ijms-22-10685-f001:**
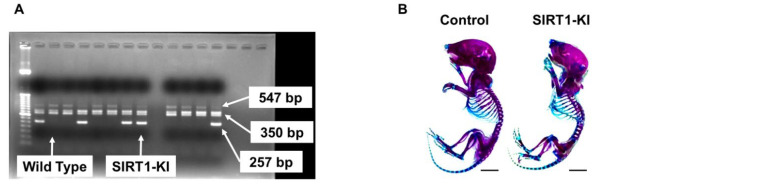
(**A**) Reverse transcription PCR. Genotyping was performed as follows: 257 bp for SIRT1-KI type; and 547 bp for wild type. (**B**) Double-staining with Alizarin red and Alcian blue of the whole skeleton of postnatal day 2 control and SIRT1-KI mice (scale bars = 5 mm). (**C**) Length of long bones and vertebra (first to fifth lumbar spines) of control (*n* = 5) and SIRT1-KI (*n* = 5) littermate embryos at postnatal day 2. NS, not significant. (**D**) Mean Body Weight (g) of Mice in the Control, SIRT1-KI at the Indicated Time Points (*n* = 5 mice/group). NS, not significant. M, male. F, female. (**E**) Sirt1 expression in control mice and SIRT1-KI mice. Relative expression level of Sirt1 mRNA was examined by real-time PCR setting the expression level in the muscle of control mice as 1 (**F**) (**a**,**b**) Safranin O-fast green staining of the medial knee joint at 3-week-old. Scale bars = 100 μm. (**c**,**d**) Hematoxylin-eosin staining of the medial knee joint at 3-week-old. Scale bars = 100 μm.

**Figure 2 ijms-22-10685-f002:**
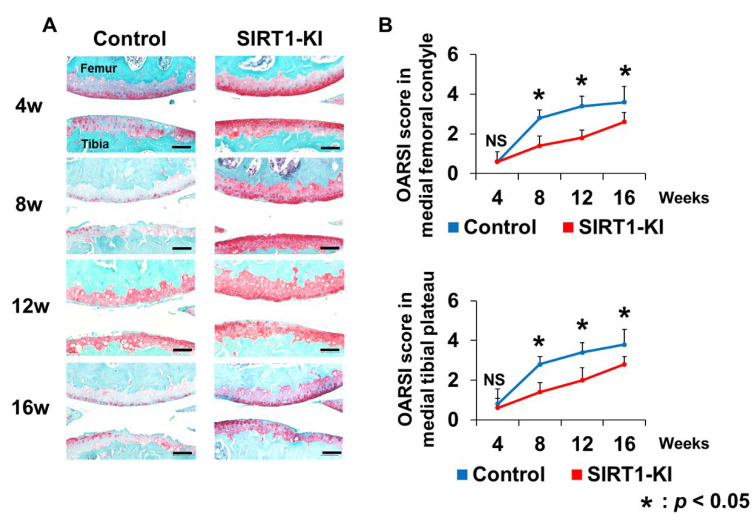
(**A**) Safranin O-fast green staining of the medial knee joint at 4, 8, 12, and 16 weeks postsurgery. Scale bars = 100 μm. (**B**) The Osteoarthritis Research Society International (OARSI) scores of the medial femoral and tibial condyle (*n* = 5 mice/group for each time point; * *p* < 0.05, NS, not significant.).

**Figure 3 ijms-22-10685-f003:**
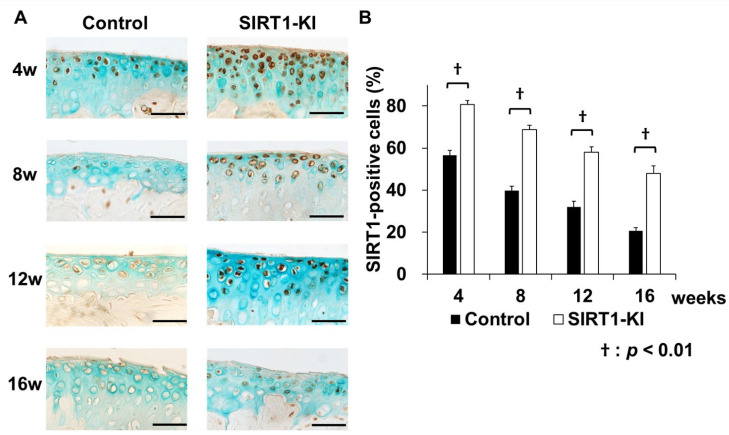
(**A**) Immunohistochemistry of SIRT1 in the medial tibial plateau at 4, 8, 12, and 16 weeks postsurgery. Scale bars = 50 μm. (**B**) The percentage of SIRT1-positive chondrocytes. Three micrographs of the medial tibial plateau were taken under ×40 magnification. The percentage was determined as the positive chondrocytes/total number of chondrocytes at ×100 magnification (*n* = 5 mice/group for each time point; ^†^
*p* < 0.01).

**Figure 4 ijms-22-10685-f004:**
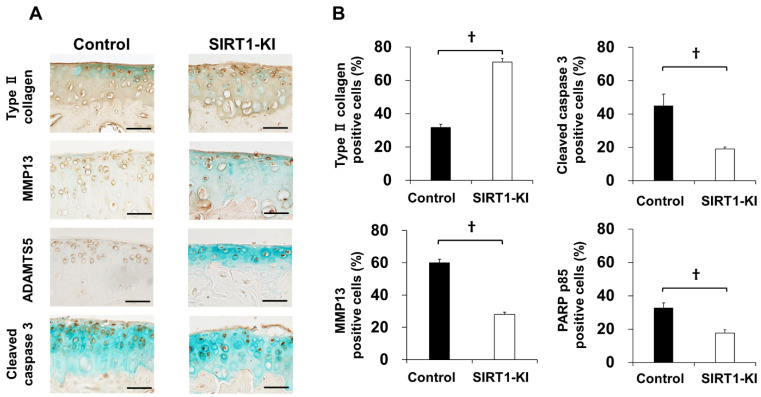
(**A**) Immunohistochemistry of type-II collagen, MMP-13, ADAMTS-5, cleaved caspase 3, PARP p85, acetylated NF-κB P65, IL-1β, and IL-6 in the medial tibial plateau at 8 weeks postsurgery. Scale bars = 50 μm. (**B**) The percentage of type-II collagen-, MMP-13-, ADAMTS-5-, cleaved caspase 3-, PARP p85 fragment-, acetylated NF-κB P65-, IL-1β-, and IL-6-positive cells. Three micrographs of the medial tibial plateau were taken at ×40 magnification. The percentage was determined as the positive cells/total number of cells at ×100 magnification (*n* = 5 mice/group for each time point; ^†^
*p* < 0.01).

**Figure 5 ijms-22-10685-f005:**
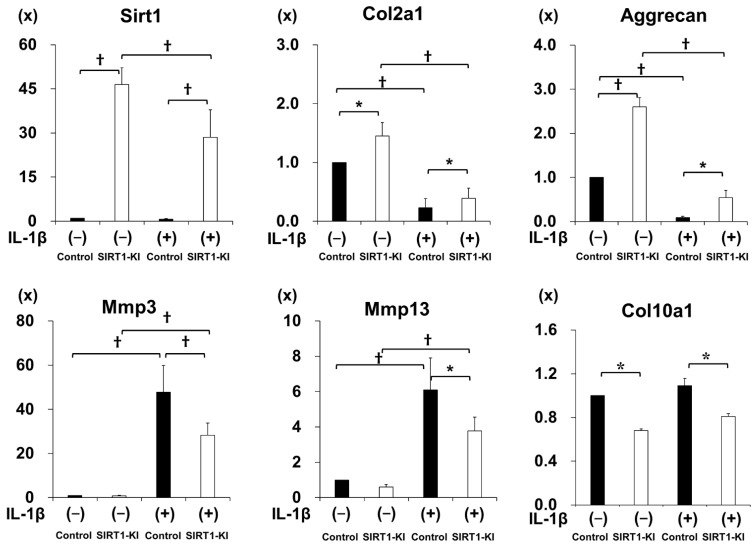
Real-time polymerase chain reaction (PCR) analysis of Sirt1, Col2a1, Col10a1, Mmp-3, Mmp-13, Aggrecan, and Adamts-5 mRNA expression in primary mouse chondrocytes. Primary mouse epiphyseal chondrocytes were cultured with 10 ng/mL IL-1β for 24 h (*n* = 5 independent experiments; * *p* < 0.05, ^†^
*p* < 0.01).

**Figure 6 ijms-22-10685-f006:**
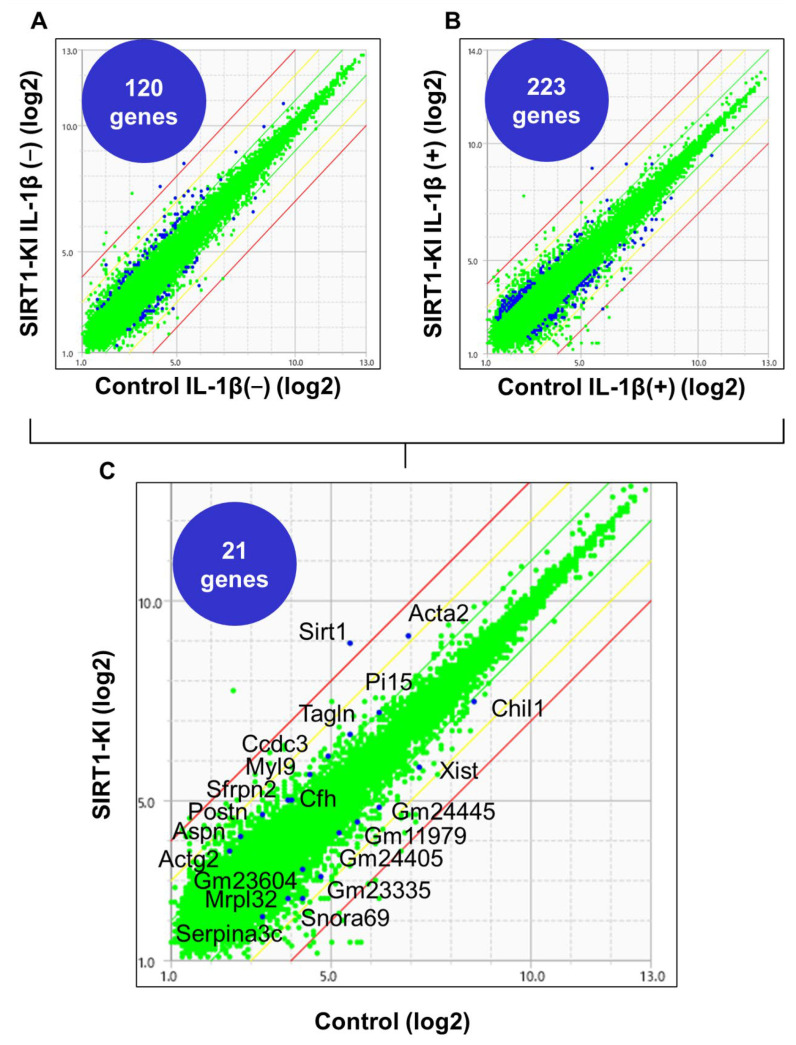
Scatter plots showing the correlation of signal values between two samples from chondrocytes without IL-1β stimulation and chondrocytes with IL-1β stimulation. (**A**) Upon gene expression profiling on SIRT1-KI chondrocytes, 120 genes were found to be upregulated or downregulated, with a two-fold change cut-off in chondrocytes without IL-1β stimulation. (**B**) Upon gene expression profiling on SIRT1-KI chondrocytes, 223 genes were found to be upregulated or downregulated, with a two-fold change cut-off in chondrocytes with IL-1β stimulation. (**C**) There were 21 genes involved in both of the above.

**Figure 7 ijms-22-10685-f007:**
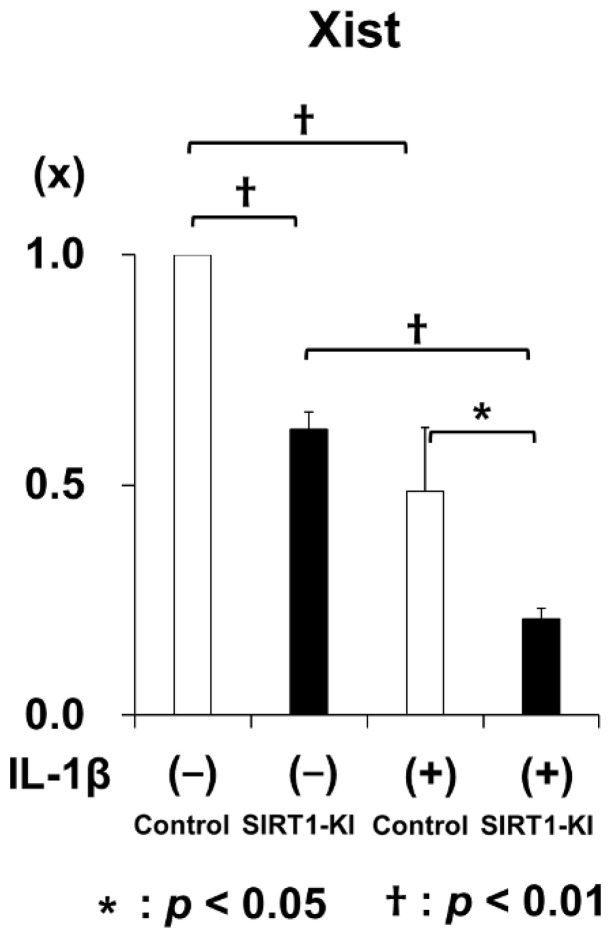
Real-time polymerase chain reaction (PCR) analysis of Xist mRNA expression in primary mouse chondrocytes. Primary mouse epiphyseal chondrocytes were cultured with 10 ng/mL IL-1β for 24 h (*n* = 5 independent experiments; * *p* < 0.05, ^†^
*p* < 0.01).

**Table 1 ijms-22-10685-t001:** Differentially regulated genes (2-fold) in control mice and SIRT1-KI mice.

No.	Genes	Log Ratio
1	Sirt1	3.484823
2	Acta2	2.169581
3	Aspn	1.389573
4	Postn	1.343505
5	Actg2	1.302517
6	Ccdc3	1.214218
7	Myl9	1.189704
8	Tagln	1.168344
9	Sfrp2	1.061471
10	Pi15	1.019975
11	Cfh	1.001788
12	Gm24405	−1.001405
13	Gm23604	−1.026104
14	Chil1	−1.117696
15	Serpina3c	−1.196805
16	Gm11979	−1.22567
17	Mrpl32	−1.343651
18	Xist	−1.368057
19	Gm24445	−1.382748
20	Gm23335	−1.61242
21	Snora69	−1.69543

**Table 2 ijms-22-10685-t002:** Primers used for real-time PCR.

Gene	Assay ID
Gapdh	Mm99999915_g1
Sirt1	Mm01168521_m1
Col2a1	Mm01309565_m1
Col10a1	Mm00487041_m1
Mmp-3	Mm00440295_m1
Mmp-13	Mm00439491_m1
Acan	Mm00545807_m1
Adamts-5	Mm00478620_m1
Xist	Mm01232884_m1

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
