# Peer review of "Knee Osteoarthritis Progression Is Delayed in Silent Information Regulator 2 Ortholog 1 Knock-in Mice"

_ijms, 2021, doi:10.3390/ijms221910685_

Round 1

Reviewer 1 Report

Yamamoto et al. investiagted, whether and how the overexpression of silent information regulator 2 ortholog 1 (SIRT1) influences the progression of experimental osteoarthritis (OA) in SIRT-knock-in mice after destabilization of medial meniscus. The authors demonstrated that the OARSI scores as well as the expression of selected catabolic markers at gene and protein level were significantly lower in SIRT1-knock-in mice compared to controls. In addition, isolated chondrocytes were treated with IL-1β. This resulted in reduced expression of anabolic (Col2a1, aggrecan) and increased expression of catabolic (MMP-3,13; ADAMTS5) markers. Also the hypertophic marker Col10a1 was induced in IL-1β-treated chondrocytes. These effects were observed in both control and SIRT1-kock-in cells, however, SIRT1-knock-in was still beneficial compared to control.

This study is of potential interest, and the findings are novel. However, some major concerns raised during review.

  1. Following two sentences (lines 11-13) are saying the same thing twice over: „Overexpression of silent information regulator 2 ortholog 1 (SIRT1) is associated with beneficial roles in aging-related diseases; however, the effects of SIRT1 overexpression on osteoarthritis (OA) progression have not yet been studied in SIRT-knock-in (KI) mice. The aim of this study was to investigate OA progression in SIRT1-KI mice using a mouse OA model.“ I would delete the marked words…

  1. Line s 13-14: the OA model was not created by the authors. Please rephrase e.g.: OA was induced via…

  1. Line 46: the introductory sentence „Studies have shown that regulation of SIRT1 may affect OA progression.“ should come earlier, before „In human chondrocytes, SIRT1 inhibits apoptosis and…“ (line 41).

  1. The authors should emphasize, at least in the discussion, why it was important to investigate SIRT1-KI mice. What is the fundamental difference to treatments with SIRT1 activators (chemicals).

  1. Could SIRT-KI have wider effects on the behavior of mice e.g. on their activity? This could influence the progression of OA.

  1. Line 76: Saf-O and Hematoxylin-eosin stainings are no immunostainings.

  1. Please indicate the Figures (e.g. Statement (Fig.1A)) in the section results. It is difficult to follow, where the reader can see the described result.

  1. 1B: the authors describe that there was no difference between control and SIRT1-KI mice regarding skeletal development. However, I do see less or delayed ossification in the ribs, fingers, vertebral bodies. This should be discussed.

  1. It is not clear in the methos section, if mice of both sexes were used for OA induction. This will be addressed at the very end of discussion fort he first time. And this is one oft he major concerns. Why did the authors use female mice? It is known that DMM is male-dominant…

  1. Another major point: Fig.2: I can not see the described progression of OA. At weeks 8-16 I can see only proteoglycane loss but no cartilage defect. This can not be valued at scores 3or 4 (when maximum is 6). Please describe the scoring system and reevaluate the results. Or, please select representative pictures for the respective graphs. In addition or consequently, I can not see the differences between SIRT1-KI and controls. They look very similar for all time points (no defect).

  1. 1. Label femur and tibia using letters.

  1. 3: It seems that controls have almost no or only very low SIRT1 expression compared to SIRT1-KI mice. This is not reflected by the graph, were controls have 60% SIRT1-positive cells and SIRT1-KI mice 80%....Pictures and graph do not fit again.

  1. 3 and 4: what is the blue staining in the cartilage matrix?

  1. 4: why was the time point 8 weeks selected fort he analyses? Why was type II collagen analyzed only in the cells and not in the matrix?

  1. 5: Why was IL-1β used in a much too high concentration? In OA, IL-1β levels are much lower in the synovial fluid (below 5 ng(ml)). This should be addressed and discussed. I disbelieve that the difference between following groups is significant: Col10a1 control with and without IL-1β, ADAMTS5 control and SIRT1-KI with IL-1β Please check.

  1. 5: How do the authors explain that IL-1β reduced the expression of SIRT1 in SIRT1-KI chondrocytes? Please discuss.

  1. Discussion: the authors state „Microarray analysis showed that 11 genes were consistently upregulated by more 222 than two-fold while 10 genes were downregulated to less than 50% of their original ex- 223 pression level in SIRT-KI chondrocytes compared to those in control chondrocytes regard- 224 less of IL-1β stimulation. However, the roles of these genes in cartilage tissues are not 225 understood at present. Therefore, whether the identified genes are involved in the regu- 226 lation associated with SIRT1 in cartilage tissues remains to be investigated“. At least 1 or 2 genes should be selected and the authors should try to discuss, how the changes in their expression could influence chondrocyte function or OA progression. Otherwise, showing the microarray results does not make any sense.

  1. Methods 4.6.: please provide a primer list.

  1. Methods 4.6 and 4.8: for the most markers it is clear why they were selected. Most researchers know, why collagens or MMPs are imprtant in this field. However, authors should briefly describe why PARP p85 or acetylated NF-κB p65 etc. were analyzed. One should consider that also researcher of other areas could read the manuscript.

Reviewer 2 Report

This is a very interesting study detailing the potentially chondroprotective role of SIRT1 in articular cartilage. However, a major concerns with this study, as a sham control has not been used during the DMM induction of osteoarthritis. Without this control it is not possible to acertain whether the upregulation of cartilage anabolism and downregulation of catabolism are stimulated by the OA induction or inherent to the SIRT1 overexpression. A sham control should be included in all the DMM experiments.

Other minor comments:

  • line 70 "the proportion of SIRT1-KI mice was approximately 40%" what crosses were these? what is the copy number of SIRT1 in these mice?
  • line 72 - Figure 1 is not referenced very well and not referred to very well in text. It does not show what the text describes, there is no immunohistochemistry on that figure.
  • Figure 1E - SIRT levels here were measured by what methods? please show a representative image and/or controls and label the units on the axis
  • line 80 - the paragraph needs more text, at what age was the meniscus destabilised and for how long? the paper needs to be bit more descriptive to allow interpretation of data
  • line 90 - "immunohistochemical analyses revealed that a considerably higher number of SIRT1-positive chondrocytes were detected in the SIRT1-KI group than that in the control group at 8, 12, and 16 weeks after surgery" this line implies that the upregulation was not only due to the overexpression of SIRT1 but also induced by surgery. The same commentary follows for the expression of type II collagen etc. A sham control is necessary to acertain which of these responses is induced by the induction of OA and which by the overexpression of SIRT1 itself.

  • line 94 - the markers that were looked at should be introduced a bit better in text
  • Figure 6 - volcano plots of the microarray data would be more informative than scatter plots. Was GO term analysis performed? Please label the axes on your graphs

Round 2

Reviewer 1 Report

All questions answered satisfactorily and the quality of the manuscript has been improved as suggested.

I have no further concerns.